

# Endophytic fungal communities associated with field-grown soybean roots and seeds in the Huang-Huai region of China

Hongjun Yang[1,2], Wenwu Ye[1,2], Jiaxin Ma[1,2], Dandan Zeng[1,2], Zhenyang Rong[1,2], Miao Xu[1,2], Yuanchao Wang[1,2] and Xiaobo Zheng[1,2]

[1] Department of Plant Pathology, Nanjing Agricultural University, Nanjing, Jiangsu Province, China
[2] The Key Laboratory of Integrated Management of Crop Diseases and Pests (Ministry of Education), Nanjing, Jiangsu Province, China

## ABSTRACT

Plants depend on beneficial interactions between roots and fungal endophytes for growth, disease suppression, and stress tolerance. In this study, we characterized the endophytic fungal communities associated with the roots and corresponding seeds of soybeans grown in the Huang-Huai region of China. For the roots, we identified 105 and 50 genera by culture-independent and culture-dependent (CD) methods, respectively, and isolated 136 fungal strains (20 genera) from the CD samples. Compared with the 52 soybean endophytic fungal genera reported in other countries, 28 of the genera we found were reported, and 90 were newly discovered. Even though *Fusarium* was the most abundant genus of fungal endophyte in every sample, soybean root samples from three cities exhibited diverse endophytic fungal communities, and the results between samples of roots and seeds were also significantly different. Together, we identified the major endophytic fungal genera in soybean roots and seeds, and revealed that the diversity of soybean endophytic fungal communities was influenced by geographical effects and tissues. The results will facilitate a better understanding of soybean–endophytic fungi interaction systems and will assist in the screening and utilization of beneficial microorganisms to promote healthy of plants such as soybean.

## INTRODUCTION

Fungal endophytes differ from pathogens, which lead to disease and reduce the fitness of their host plants. Fungal endophytes inhabit the asymptomatic aboveground and underground tissues of their hosts and are found in all species and in all divisions of land plants (*Zimmerman & Vitousek, 2012*). Endophytes often form mutualistic interactions with their host, with the relationship benefitting both partners (*Deshmukh et al., 2006*). Fungal endophytes and land plants have interacted for over 400 million years (*Krings et al., 2007*). Since Vogl isolated and cultured symptomless endophytes from seeds of *Lolium temulentum* (*Vogl, 1989*), fungal endophytes have been studied in many

Corresponding authors
Wenwu Ye, yeww@njau.edu.cn
Xiaobo Zheng,
xbzheng@njau.edu.cn

systems, including the diversity and function of clavicipitaceous and non-clavicipitaceous fungal endophytes that infect grasses, nonvascular plants, ferns and allies, conifers, and angiosperms (*Clay, 1989*; *Rodriguez et al., 2009*). In addition, the relationships between endophytic fungi and medicinal plants have also been reported (*Jia et al., 2016*).

Endophytes have established long-lasting interactions with different hosts, involving growth stimulation, alleviation of salt stress, and induction of local and systemic resistance to pathogens (*Lahrmann et al., 2013*). For example, the endophytic fungus, *Phomopsis* sp., produces piperine, indole-3-acetic acid, and gibberellic acid to promote plant growth (*Chithra et al., 2017*), endophytic fungi isolated from the roots of *Sophora tonkinensis* Gapnep inhibit three fungal phytopathogens of *Panax notoginseng* (*Yao et al., 2017*), and endophytic *Penicillium* strains produce gibberellins to overcome salt stress (*Leitao & Enguita, 2016*). In addition, the biosynthetic products of the endophytic fungus *Nigrospora oryzae*, isolated from *Combretum dolichopetalum* leaves, possess strong antidiabetic activity (*Uzor, Osadebe & Nwodo, 2017*). Endophytic fungi are therefore a source of novel compounds promoting organic farming.

In comparison with other crops, studies on endophytes in soybean plants are still limited, especially those grown in China, a country with a long history of soybean growing. In Brazil, 12 genera of fungal endophytes were isolated from the leaves and stems of soybeans. There was no difference in the number of fungi isolated from the leaves and stems; however, the number of endophytes decreased as the plants aged, and more fungi were found in tissues near the soil (*Pimentel et al., 2006*). In Argentina, 11 genera of fungal endophytes were identified in soybean leaves, stems, and roots. Fungal endophytic colonization in soybean plants was influenced by tissue type, with a greater number of endophytes isolated from stems than from leaves and roots (*Russo et al., 2016*). In the U.S., endophytes in the stems of soybean plants were isolated using culture-dependent (CD) and culture-independent (CI) colony isolation methods, with 12 and six genera identified, respectively (*Impullitti & Malvick, 2013*). Many of the fungal endophytes identified in the above studies are known to be soybean pathogens, e.g., *Fusarium* and *Alternaria*; however, they were not able to cause symptoms of disease. It remains to be determined whether those endophytes are latent pathogens or non-pathogenic forms that benefit the host.

For the characterization of endophytic fungal communities, traditional methods of species identification such as morphological observations and sequencing of PCR products to identify molecular marker(s) have been routinely used in previous studies. However, several molecular technologies, e.g., terminal-restriction fragment length polymorphism (T-RFLP), denaturing gradient gel electrophoresis (DGGE), and metagenomics, are now being used to better characterize endophytes and their roles in plant ecophysiology (*Rodriguez et al., 2009*). For example, DGGE was used to characterize the diversity and phylogenetic relationships of endophytic fungi in *Bletilla ochracea* (*Tao et al., 2008*), T-RFLP was used to analyze the structure of ectophytic potato-associated bacterial communities and antagonistic behaviors toward plant pathogenic fungi (*Berg et al., 2005*), and a metagenomics approach was used to analyze an endophytic bacterial community in the roots of rice (*Sessitsch et al., 2012*).

In the present study, we characterized the endophytic fungal communities associated with field-grown soybean roots and corresponding seeds in the Huang-Huai region of China, which is one of the most important soybean producing areas. We used both CD and CI methods for colony isolation and sequencing, including PCR sequencing and internal transcribed spacer (ITS) high-throughput sequencing to identify species. The results from different methods, different countries and cities in China, and different tissues (roots and seeds) were compared to evaluate the diversity of endophytic fungal communities associated with soybeans.

## MATERIALS AND METHODS

### Field trials and sampling

A total of six fields in three cities of the Huang-Huai region of China, Jining of Shandong Province (35°27′N, 116°35′E), Xuzhou of Jiangsu Province (34°17′N, 117°17′E), and Suzhou of Anhui Province (33°38′N, 117°05′E), were selected for this study; the corresponding samples were named JN1 and JN2, XZ1 and XZ2, and SZ1 and SZ2, respectively. The two fields in each city were adjacent, and each field was 30 × 8 m in size. The fields had been under a similar agronomic management and fertilization regime for three years. The crop planting pattern was a soybean–wheat rotation, and the soybean cultivar was Zhong Huang 13, which is widely grown in Huang-Huai region of China.

The seeds to be grown were harvested from the previous year; those harvested in 2015 and sown in 2016 were sampled for assays, with a suffix "a" at the end of the sample name. The seeds were stored at 4 °C until analyzed. Healthy adult soybean plants were sampled during July and September of 2016. Each sampling was comprised of 15 soybean plants collected in the shape of a W from 15 sites in each field. The time points for sampling were 30, 60, and 90 days after seed sowing, which corresponded to the flowering, pod formation, and maturation stages of the soybean growth period, with suffixes of "b," "c," and "d" at the end of the sample names, respectively. The loosely adhering soil on the roots was removed by shaking, immediately placed into sterile plastic bags in an ice box, and then transported to the laboratory for analyses.

### Pretreatment of the samples

After a washing step to remove soil residue and dust, the roots of each plant were cut into segments of 5–7 cm, and 2 g of each root segment was analyzed. Fifteen plants collected from the same field were pooled, and 10 g of the pooled root segment sample were selected randomly. The 10 g root segments were supplemented with 200 mL distilled water and three drops of Tween 20 at 25 °C, then shaken at 220 rpm for 20 min. We also randomly selected 10 g of the seed sample for analyses. Surface sterilization was performed following previous protocols (*Impullitti & Malvick, 2013*; *Potshangbam et al., 2017*) with some modifications; each root and seed was successively washed with sterile water for 20 s, immersed in 70% ethanol for 5 s, soaked in 0.525% sodium hypochlorite for 2 min, washed with sterile distilled water again three or four times, then dried under sterile conditions. Each sample was divided into two parts, which were used for the CD and CI methods, as described below.

**Table 1 The primers used in this study.**

| Assay | Target gene (region) | Primer | Sequence (5′–3′) |
|---|---|---|---|
| Regular PCR | ITS1, 5.8S, and ITS2 | ITS1-F | CTTGGTCATTTAGAGGAAGTAA |
| | | ITS4 | TCCTCCGCTTATTGATATGC |
| Regular PCR | EF-1α | EF-1 | ATGGGTAAGARGACAAGAC |
| | | EF-2 | GGARGTACCAGTSATCATGTT |
| Amplicon sequencing | ITS1 | ITS1-F | CTTGGTCATTTAGAGGAAGTAA |
| | | ITS2 | GCTGCGTTCTTCATCGATGC |

## The CD method

The CD method was used to obtain pure fungal colonies for further identification of individual species and to obtain mixed fungal colonies for ITS sequencing. The root segments were further cut into shorter segments (0.25 cm), and every five randomly selected segment was placed on a plate containing potato dextrose agar (PDA), malt extract agar, synthetic potato medium, and czapek dox medium. All culture media were also amended with ampicillin (50 mg/L) and rifampicin (50 mg/L) to inhibit bacterial growth. There were a total of 360 pieces (five segments per plate × three plates × four culture media × six fields), which were maintained at 25 °C.

To identify the species of pure fungal colonies, after 3–5 days, when mycelia emerged from the root tissues, small pieces of medium together with mycelia from the margins of growing cultures were transferred to a new PDA plate. These pure fungal cultures were preliminarily grouped into morphotaxa, based on mycelium type, colony color, and growth rate. After seven days of growth, mycelia DNA was extracted using the DNAsecure Plant Kit (Tiangen, Beijing, China) according to the manufacturer's instructions. Nuclear rDNA regions, including ITS1, 5.8S, and ITS2, were amplified by PCR in a 30 μL reaction containing 15 μL 2× EasyTaq PCR SuperMix (+dye) (TransGen, Beijing, China), 1 μL each of the primers ITS1-F and ITS4 (Table 1), 1 μL template DNA solution, and 12 μL sterile water, using previously described reaction parameters (*Liang et al., 2014*). For *Fusarium* species, we also used primers EF-1 and EF-2 to amplify the translation elongation factor (EF)-1α region for confirmation (*O'Donnell et al., 2010*) (Table 1). The PCR products were sequenced, and the sequences were compared with the NCBI database (http://www.ncbi.nlm.nih.gov; using the BLASTN program) for identification of species.

For ITS sequencing of mixed fungal colonies, after three weeks of fungal growth in root segments, the mycelia on the surface of the culture medium were scraped, and those from the same type of culture medium were pooled for DNA extraction. Equal volume of DNA from mycelia grown on four different kinds of culture medium were mixed into one sample for ITS sequencing as described below.

## The CI method

A total of 10 g of the root or seed samples were frozen and ground in liquid nitrogen using a mortar and pestle. Approximately 400 mg each sample were used to extract genomic DNA according to the method described above.

## MiSeq sequencing and data analyses

Internal transcribed spacer sequencing for identification of fungi was performed following a previous protocol (*Kozich et al., 2013*). The ITS1 region of the fungal ITS was amplified using the ITS1-F and ITS2 primers (Table 1). After successful amplification, the PCR products were purified using Ampure XP beads (Agencourt Bioscience, Beverly, MA, USA) to remove nonspecific products. The libraries were evaluated as follows: (1) the average molecule length was determined using the Agilent 2100 Bioanalyzer and Agilent DNA 1000 kit (Agilent Technologies, Santa Clara, CA, USA), and (2) quantitative real-time PCR using EVAGreen™ (Jena Bioscience, Jena, Germany) was used to quantitate the libraries. The final libraries were sequenced using the MiSeq system and the MiSeq reagent kit (Illumina, San Diego, CA, USA) using PE250 (Illumina, San Diego, CA, USA).

The raw sequence reads were first filtered according to the default parameters. The overlapping clean reads were assembled into consensus sequences (namely tags) using the Fast Length Adjustment of SHort reads, version 1.2.11 (http://ccb.jhu.edu/software/FLASH) program. The tags were further clustered into operational taxonomic units (OTUs) with a sequence identity of 97% as the threshold using UPARSE (http://www.drive5.com/uparse), and chimeras were filtered out using UCHIME, version 4.2.40 (http://drive5.com/uchime). The OTUs were taxonomically classified using Ribosomal Database Project Classifier, version 2.2 (http://rdp.cme.msu.edu), referencing the UNITE ITS database (https://unite.ut.ee) using 0.6 confidence values as the cutoff.

## Statistical analysis

We performed α-diversity and principal coordinate analyses using R package (version 3.1.1; https://www.r-project.org). Data from the different bioassays were compared by one-way analysis of variance using SPSS Statistics 20 software for Windows (SPSS, Chicago, IL, USA). The Pearson and Spearman correlations for data from selected taxonomic groups (phyla or genera) were also calculated using this software.

# RESULTS

## Endophytic fungal communities in the collected healthy soybean root samples

We analyzed endophytic fungal communities by high-throughput ITS sequencing of 18 CI root samples (CI-seq), e.g., the six samples from Jining were designated as JN1b, JN1c, JN1d, JN2b, JN2c, and JN2d. The resulting high-quality reads with >97% sequence identity were clustered into 385 OTUs, ranging from 37 to 116 OTUs per sample. Annotation of these OTUs resulted in identification of 105 fungal genera (Table S1). We also analyzed 18 CD root samples by high-throughput ITS sequencing (CD-seq) and identified a total of 50 fungal genera (Table S1). In addition, 136 fungal strains belonging to 20 genera were isolated from the CD root samples (CD-iso; Table S2). Thus according to the number of genera, the throughputs of the three methods ranked as CI-seq > CD-seq > CD-iso.

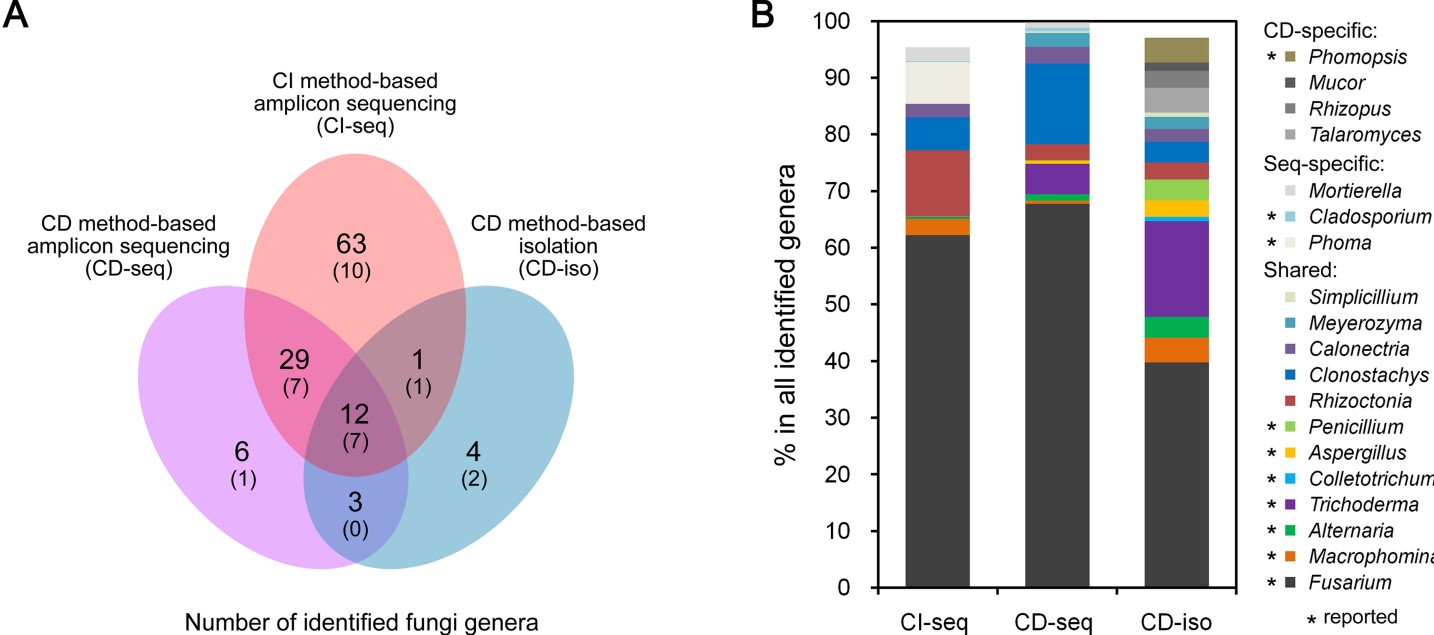

**Figure 1 Root-associated endophytic fungi detected using three different methods.** (A) Number of endophytic fungal genera detected using different methods. The numbers of previously reported endophytic fungal genera were also indicated in parentheses. (B) The relative abundances of the indicated endophytic fungal genera. Culture dependent-specific fungal genera were only identified using CD-seq and/or CD-iso, and sequencing-specific fungal genera were only identified using CI-seq and/or CD-seq. Shared fungal genera were identified by all three methods. "*," fungal genera that have been reported as endophytic fungi.

Among all identified 142 genera, only 98 were identified using the high-throughput ITS sequencing method (CI-seq and CD-seq) rather than CD-iso. The OTU abundances of the 98 genera were 14.6% by CI-seq and 1.8% by CD-seq. Twelve genera were identified by the CD method only (CD-seq and CD-iso) rather than CI-seq. The OTU abundances were 0.2% by CD-seq and 15.4% by CD-iso.

Twelve genera were identified by all three methods (Fig. 1A). The OTU abundances of these 12 genera were over 80% (85.4%, 98.2%, and 83.8%) of all genera identified by CI-seq, CD-seq, and CD-iso, respectively (Table S1). Among the 12 genera, *Fusarium* exhibited the highest abundance using all three methods; the OTU abundances of *Fusarium* were over half using CI-seq (62.2%) and CD-seq (67.8%), and 39.7% using CD-iso (Fig. 1B; Table S1). The abundances of the other genera were much lower than *Fusarium* and varied depending on the method of analysis. For example, the second most abundant genera using CI-seq, CD-seq, and CD-iso were *Rhizoctonia* (11.7%), *Clonostachys* (14.3%), and *Trichoderma* (16.9%), respectively (Fig. 1B; Table S1).

## Geographical effects on soybean root endophytic fungal communities

We further analyzed the CI-seq results of samples from three different cities to determine if geographical location affected the root endophytic fungal communities. We identified 77, 80, and 75 fungal genera from Jining, Suzhou, and Xuzhou, respectively, and 25, 28, and 23 genera, respectively, were not identified in at least one other city

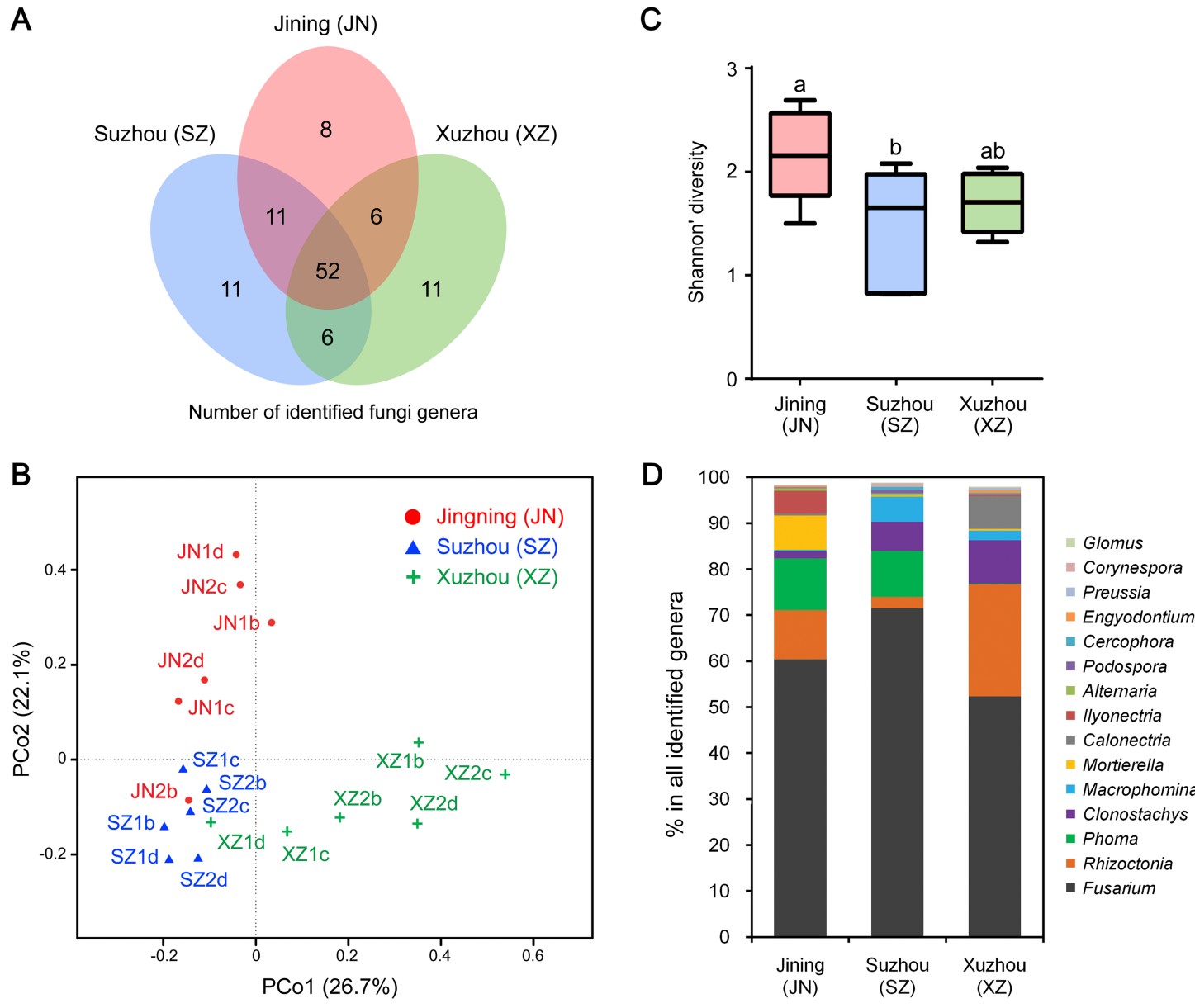

**Figure 2 Root-associated endophytic fungi identified at three locations.** (A) The number of endophytic fungal genera detected from three locations. (B) The results of principal coordinate analyses; every sample is labeled with a specific color to designate the location. (C) Within-sample diversity (α-diversity or Shannon's diversity) analyses of root samples. The horizontal bars within boxes represent the medians. The upper and lower edges of the boxes represent the 75th and 25th quartiles, respectively. The upper and lower whiskers extend 1.5-fold of the interquartile range from the upper and lower edges of the box. (D) The relative abundances (%) of the indicated endophytic fungal genera from three locations.

(Fig. 2A; Table S3). However, the OTU abundances of these non-core genera were only 0.4%, 0.6%, and 0.3% of all genera in the corresponding cities, respectively (Table S3).

Principal coordinate analyses of weighted distances were performed to investigate patterns of separation between microbial communities. The samples from Xuzhou were separated from those of Jining and Suzhou along the first principal coordinate,

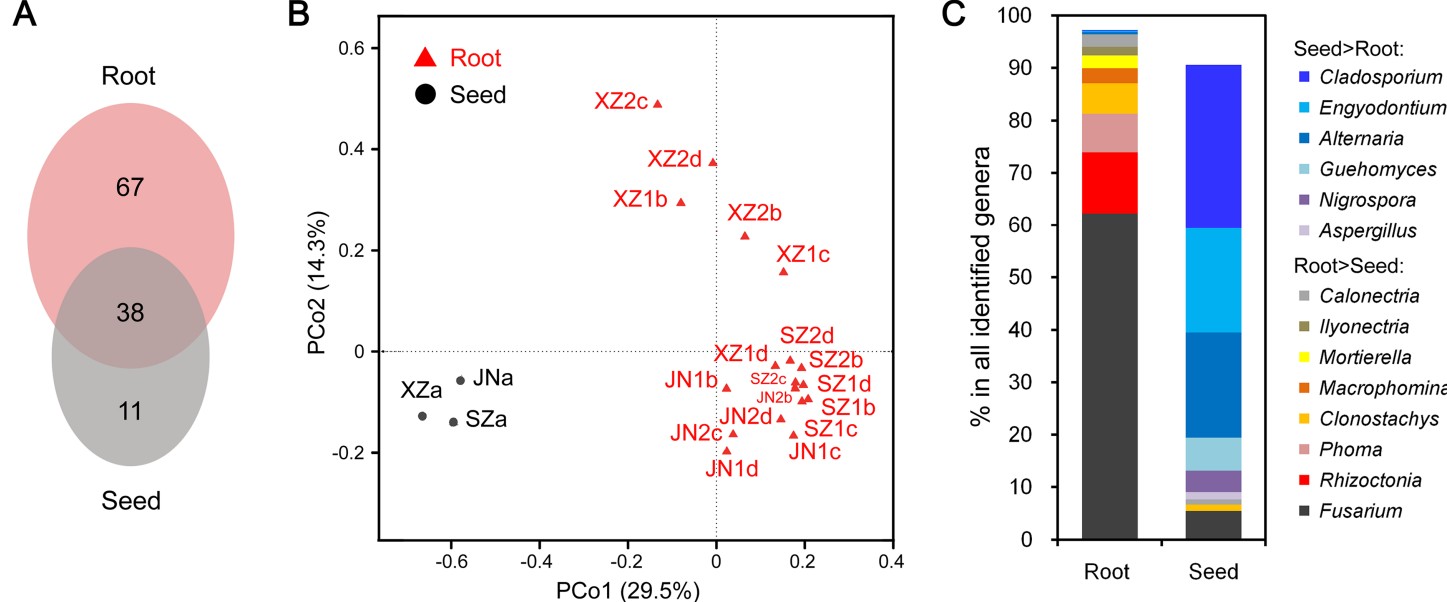

**Figure 3 Endophytic fungi detected in seeds and roots of soybeans.** (A) Numbers of endophytic fungal genera detected from root and seed samples. (B) The results of principal coordinate analyses; every sample is labeled with a specific color to designate the tissue. (C) Relative abundances (%) of the indicated endophytic fungal genera in the root and seed samples.

PCo1, and the samples of Jining were separated from the others along the principal coordinate, PCo2, indicating that endophytic fungal communities varied among the different cultivation sites (Fig. 2B). Analyses of within-sample diversity (α-diversity or Shannon's diversity) showed that endophytic fungal communities of Jining had the highest diversity, in contrast to samples from Suzhou that were significantly different (Fig. 2C).

The geographical effects on the soybean root endophytic fungal communities were present in the dominant core genera. For example, there were 52 fungal genera shared in all three cities, and the top five most frequent genera were *Fusarium*, *Rhizoctonia*, *Phoma*, *Clonostachys*, and *Macrophomina*. Among the results from three cities, Suzhou had the highest OTU abundance in *Fusarium* and *Macrophomina*, Xuzhou had the highest OTU abundance in *Rhizoctonia*, *Clonostachys*, and *Calonectria*, and the lowest OTU abundance in *Phoma* (Fig. 2D; Table S3).

## Significant differences in endophytic fungal communities between roots and seeds

The endophytic fungal species detected in plants could be influenced by many factors, such as the type of tissue. We further compared the CI-seq results of endophytic fungal communities between the sampled roots and corresponding seeds for soybean plant growth. There were 38 common fungal genera between the root and seed samples, with 67 genera unique in roots, while only 11 genera were unique in seeds (Fig. 3A; Table S4). Using principal coordinate analyses, the root and seed samples were clearly distinct along the first principal coordinate (PCo1; Fig. 3B).

The most abundant annotated genera in roots were *Fusarium*, *Rhizoctonia*, *Phoma*, *Clonostachys*, *Mortierella*, and *Calonectria*, and the OTU abundances of these genera were all higher than those in seeds. Genera such as *Macrophomina* and *Ilyonectria* were identified in roots only (Fig. 3C; Table S4). In contrast, *Cladosporium*, *Alternaria*, *Engyodontium*, *Guehomyces*, *Nigrospora*, and *Aspergillus* were more abundant in seeds than in roots (Fig. 3C; Table S4).

## DISCUSSION

To the best of our knowledge, this is the first report of fungal endophytic diversity in field-grown soybean roots in the Huang-Huai region of China. Three methods of analyses were used to obtain a complete qualitative picture of endophytic fungal communities. Compared with the CD-iso method, the high-throughput sequencing-based CD-seq and CI-seq methods showed significant advantages in terms of sensitivity. The CI-seq method allows detection and identification of all fungi including those that are cryptic, sterile, and unculturable, and potentially detects a more diverse endophyte population than CD-iso methods (*Impullitti & Malvick, 2013*). However, the CD-seq and CI-seq methods amplified shorter DNA fragment and their identifications were subsequently compromised to only genus-level. Compared with the CI-seq method, CD-iso method recovered the lowest quantity of fungal population, but its identification had the best quality. Most fungal isolates were identified to species-level (Table S2). In addition, CD-iso method could obtain fungal culture, and provides fundamental materials for studying the biological and ecological roles in subsequent research. Interestingly, some fungi were only found by isolation, and the reason has not yet been explained.

Of 52 previously reported endophytic fungal genera, 28 (53.8%) were identified in our study, and 25 (89.2%; 25/28) of these genera were identified using CI-seq (Fig. 1A; Table S1). The remaining 24 of the 52 reported genera were not identified by any method in our study (Table S1), possibly because the analyzed samples were collected from different tissues such as leaves or stems, and the soybean plants were different cultivars and grown under different environments. There were 90 genera newly identified in our study. Genera such as *Rhizoctonia* and *Clonostachys*, which showed high OTU abundances by high-throughput ITS sequencing (CI-seq and CD-seq) but low by CD-iso, have not been reported previously (Fig. 1B; Table S1), possibly because previous studies used CD-iso- or CD-iso-like methods, rather than high-throughput methods. *Fusarium* was also reported as endophytic fungi in soybean roots and leaves, further revealing that it is a dominant fungi species in soybean endophytic fungal community.

Many soybean pathogens colonize roots, and the caused root diseases lead to extensive losses in soybean production worldwide. Over the long period of coexistence and evolution, different relationships have been established between endophytic fungi and their host plants through specific fungus–host interactions identified as a continuum of mutualism, antagonism, and neutralism (*Jia et al., 2016*). It is therefore valuable not only to identify the pathogens in disease soybean roots but also to systematically identify the microbial communities in healthy roots. We found many fungal genera/species in healthy

soybean roots that are well-known pathogen species of soybean and/or other plants such as wheat. For example, *Fusarium*, *Macrophomina*, *Rhizoctonia*, and *Alternaria* cause root rot or foliar and stem blight in soybeans (*Lu et al., 2015a*, *2015b*; *Ramos et al., 2016*; *Yamamoto et al., 2017*). Our preliminary pathogenicity assays for some of these isolates showed that many of them may be truly asymptomatic in soybean seedlings, while some were able to cause disease symptoms, indicating that they may be latent pathogens that cause disease under stress or at a later time during plant growth. Other identified endophytic fungal genera, such as *Ceratobasidium* and *Phoma*, have rarely been associated with soybean diseases (*Impullitti & Malvick, 2013*). However, these genera have been reported to be major pathogens in other plants such as wheat (*Kriuchkova, 2013*; *Perello & Moreno, 2005*). Among the endophytic fungi isolated from soybean, a high dominance was observed in the roots of *Fusarium*, and *Fusarium* has been isolated from various parts of the soybean plant (*Pimentel et al., 2006*; *Russo et al., 2016*; *Impullitti & Malvick, 2013*), which can be worrisome because it undetermined whether these isolates are non-pathogenic and beneficial to the plant or if they are latent pathogens. Overall, the significance of endophytism of endophytes in soybeans is still largely unknown and merits further investigation.

The endophytic fungal communities of soybeans may be influenced by many factors, such as the type of tissue sampled and the location in which they are grown. Greater numbers of endophytes were isolated from stem tissues than from leaves and root tissues in both soybeans and corn (*Russo et al., 2016*). In the medicinal plant *Kadsura angustifolia*, the fungal species diversity differed significantly between stems and roots, with roots containing a greater diversity than stems (*Huang et al., 2015*). There were significant differences in the endophytic communities among different regions of the Great Lakes Basin (*Clay et al., 2016*). Another study reported significant site and location variations in fungal endophytic communities (*Zimmerman & Vitousek, 2012*). In our study, we found that soybean roots and seeds harbored different endophytic fungal communities. The numbers of fungal genera were two-fold greater in roots than in seeds, while the number of root-specific fungal genera were six-fold greater than that of seed-specific fungal genera, suggesting that the field environment, such as the soil conditions, affected the endophytic fungal communities of soybean plants after the seeds have been sown. The root specific fungi might initially be present in the soil, followed by colonization of the roots and the entire plant after seed germination. Because of different soil properties among the three cities (JN, a sandy loam soil; XZ, a yellow loam sand soil; SZ, a mortar black soil), the endophytic fungal communities were also diverse. Soil type and growth conditions also shape fungal communities inhabiting the *Arabis alpina* root endosphere, suggesting that environments and soils are the main drivers of fungal alpha diversity, and the plant growing condition is the main factor structuring root-associated fungal communities, i.e., determining the taxa present and their abundances (*Almario et al., 2017*). Thus the specific impact of each environment factor still needs to be identified in future studies.

## CONCLUSION

In this study, we characterized the endophytic fungal communities associated with the roots and corresponding seeds of soybeans grown in the Huang-Huai region of China. We identified the major endophytic fungal genera of soybeans and revealed that the diversity of soybean endophytic fungal communities was influenced by geographical effects and tissues. The results will facilitate a better understanding of soybean–endophytic fungi interaction systems and will assist in the screening and utilization of beneficial microorganisms to promote healthy of plants such as soybean.

### Funding

This work was supported by grants to Xiaobo Zheng and Yuanchao Wang from the China Agriculture Research System (CARS-004-PS14), and by grants to Wenwu Ye from the National Natural Science Foundation of China (31772140). The funders had no role in study design, data collection and analysis, decision to publish, or preparation of the manuscript.

### Grant Disclosures

The following grant information was disclosed by the authors:
China Agriculture Research System: CARS-004-PS14.
National Natural Science Foundation of China: 31772140.

### Competing Interests

The authors declare that they have no competing interests.

### Author Contributions

- Hongjun Yang performed the experiments, analyzed the data, contributed reagents/materials/analysis tools, prepared figures and/or tables, authored or reviewed drafts of the paper, approved the final draft.
- Wenwu Ye conceived and designed the experiments, analyzed the data, contributed reagents/materials/analysis tools, prepared figures and/or tables, authored or reviewed drafts of the paper, approved the final draft.
- Jiaxin Ma performed the experiments, contributed reagents/materials/analysis tools, approved the final draft.
- Dandan Zeng contributed reagents/materials/analysis tools, approved the final draft.
- Zhenyang Rong contributed reagents/materials/analysis tools, approved the final draft.
- Miao Xu contributed reagents/materials/analysis tools, approved the final draft.
- Yuanchao Wang conceived and designed the experiments, authored or reviewed drafts of the paper, approved the final draft.
- Xiaobo Zheng conceived and designed the experiments, approved the final draft.

## Data Availability

The raw data are provided in the Supplemental File.

## Supplemental Information

Supplemental information for this article can be found online at http://dx.doi.org/10.7717/peerj.4713#supplemental-information.

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
