# Peer review of "Endophytic fungal communities associated with field-grown soybean roots and seeds in the Huang-Huai region of China"

_PeerJ, doi:10.7717/peerj.4713_

## Round 0.1 · original submission · Minor Revisions

I agree with Rev#2 that the Discussion section needs to be strengthened and this could be made e.g. by slightly restructuring Results and Discussion sections following the detailed suggestions of the reviewer.

Reviewer 1 ·

Basic reporting

This manuscript reports a straight forward study of fungal diversity in soybean roots. It is clear and unambiguous in describing the study and results. Background literature is sufficient. The figures are well made and clear. It is self contained.

Experimental design

The experimental design is fine. This was a survey of fungi in roots and seeds. The authors do indicate how this work will contribute to further studies on use or role of endophytes in agriculture. The study seems well done to assess diversity of fungi in soybean. The described methods were adequate.

Validity of the findings

These data are robust and valid--and conclusions support by results.

Additional comments

This study of diversity adds to our knowledge of diversity of fungal endophytes in plants. A next step would be to use the isolated endophytes to assess affects on soybean. This is really where this research would go in the future. This is a nice foundation.

·

Basic reporting

no comment

Experimental design

no comment

Validity of the findings

no comment

Additional comments

This manuscript was generally well written. The study was well designed. Specifically, using both conventional (cultural isolation followed by conventional Sanger sequencing) and innovative (Illumina MiSeq sequencing using eDNA) to study fungal diversity is the best merit of this study. The results were properly analyzed. Comparisons among results obtained from three methods plus previous reports have important implications. Here are some comments in order to help the authors to further improve the manuscript.

Abstract
The Abstract reads fine, while I encourage the authors to further improve it. First, there are already many numbers in the Abstract, which is somehow confusing. Are the numbers “18” in line 22 and 23 necessary? Second, is “based on internal transcribed spacer (ITS) sequencing” in line 23-24 necessary? Third, the sentence in line 25-26 is confusing and can be improved. Fourth, in line 27-29, although the analyses indicated the difference in population and diversity, it is confusing in stating “significantly different” after indicating “Fusarium……in EVERY sample”. Please improve the logical flow, for example, add “even though”, or “except for” before the “Fusarium in every sample” statement. Fifth, the statement in line 29 is not true. Phyllosphere fungal endophytes were not investigated in this study.

Introduction
The Introduction generally reads fine. To better serve the purpose of literature review and correlate with the later findings, I suggest the authors add literature reviews on the major fungal genera found in this study. Specifically, previous reports of Fusarium as endophytes should be reviewed, since it was the dominant genus found in this study.

Line 70-77: T-RFLP and DGGE are not “new” or “emerging”.

Materials and Methods
Line 95: the authors may want to add a sentence indicating the popularity of the ZhongHuang 13 cultivar in the Huang-Huai region to justify this selection.
Line 100: “in the field” or “in each field”?
Line 124: “Amended” to replace “treated”
Line 126: delete “kinds of”; replace “3 cities x 2” with “6”
Line 130: delete “carefully”
Line 133 to 141: shorten this section if the methods were the same as those in cited articles.
Line 144: What method was used to quantify DNAs if “equal amounts of DNA…were mixed”?
Line 153: Revise the section title to specify that this section was about sequencing and sequence analyses of the MiSeq or CI.
Line 173-176: indicate where and how the diversity analyses and subsequent graphics plotting were conducted.

Results
The Results section is generally clean and well written. The findings are clearly reported. The figures are nicely plotted/made. Minor mistakes are still present. The major issue is that many discussions should be moved to the next section.
Line 189: The term “ITS sequencing method” is misleading. ITS sequencing was also used in the CD-iso method following cultural isolation. The authors may want to find another way to describe these methods.
Line 191-192: “…indicating that…..of low abundance” belongs to the Discussion. Please only report findings in the Results, while move all related discussions to the Discussion. Also, this statement of indication is contradictory to that in Line 270-272.
Line 194-196: “Thus, …..” belongs to the Discussion.
Line 199-200: “indicating……” belongs to the Discussion.
Line 206-208: “These difference might……” belongs to the Discussion.
Line 210-221: Please only report findings from this study. This whole section should be moved to the Discussion.
Line 251-257: “These results suggested that……seed germination” and “further indicating that…..” should be moved to the Discussion. Please simply report the findings.

Discussion
Compared to the above sections, the Discussion section feels weak. It should be strengthened after moving many discussions in the Results to the Discussion.

Line 266-274: The best merit of the methodology in this study was using a variety of isolation/identification methods. The results from different methods were well illustrated in the Results. However, they deserve better comparison and discussion here. There is a lot that can and should be discussed. For example, obtaining fungal cultures provides fundamental materials for studying the biological and ecological roles in subsequent research. This is also related to the following discussion paragraph on the determination of endophytisms and pathogenicity of the isolated fungi. Second, CD-iso recovered the lowest quantity of fungal population, but its identification had the best quality. Most fungal isolates were identified to species-level with >98% identity (Table S2). On the other hand, the other methods amplified shorter DNA fragment and their identifications were subsequently compromised to only genus-level. Third, it is interesting that some fungi were only found by isolation. Was the less abundance the only reason, or were there other potential reasons related to the isolation and sequencing methods?

Line 292: The dominance and potential role of Fusarium deserve a more detailed discussion.
Line 293-308: More in-depth discussions are warranted in this paragraph. Was there any difference in the soil types and weather conditions at the time samples were collected among these fields? The discussion here can be more valuable if such potential correlation between fungal community and environmental conditions are discussed.
Also, this may be a good place to indicate that phyllosphere fungal endophytes were not studied and warrant future research.

---

## Round 0.2 · accepted · Accept

I think that in the current version of the manuscript all the reviewers' remarks have been successfully addressed. Therefore the ms is now suitable for publication in PeerJ.

#